# Efficacy and Safety of Renal Function on Edoxaban Versus Warfarin for Atrial Fibrillation: A Systematic Review and Meta-Analysis

**DOI:** 10.3390/medicines10010013

**Published:** 2023-01-16

**Authors:** Yapeng Wang, Li Li, Zhanlan Wei, Shan Lu, Wenxue Liu, Janghui Zhang, Junbo Feng, Dongjin Wang

**Affiliations:** 1Department of Cardio-Thoracic Surgery, Nanjing Drum Tower Hospital, Chinese Academy of Medical Sciences & Peking Union Medical College, Peking Union Medical College Graduate School, Nanjing 210008, China; 2Department of Cardiovascular Surgery, The First Affiliated Hospital of Anhui Medical University, Hefei 230022, China; 3Department of Thoracic and Cardiovascular Surgery, Affiliated Drum Tower Hospital of Nanjing University Medical School, Institute of Cardiothoracic Vascular Disease, Nanjing University, Nanjing 210008, China; 4Instructional Technology and Media, Columbia University, New York, NY 10027, USA; 5Department of Epidemiology and Biostatistics, School of Public Health, Anhui Medical University, 81 Meishan Road, Hefei 230032, China; 6Department of Cardio–Thoracic Surgery, Nanjing Drum Tower Hospital, Nanjing Medical University, Nanjing 210029, China; 7Nanjing Drum Tower Hospital, Xuzhou Medical University, Xuzhou 221004, China; 8Department of Cardio–Thoracic Surgery, Nanjing Drum Tower Hospital, The Affiliated Hospital of Nanjing University Medical School, Nanjing 210011, China

**Keywords:** atrial fibrillation, edoxaban, creatinine clearance, vitamin K antagonist

## Abstract

**Background**: Edoxaban is a novel oral anticoagulant which may decrease the risk of stroke and systemic embolism in patients suffering from atrial fibrillation (AF). However, the decreased efficacy of edoxaban versus warfarin for the avoidance of stroke and systemic embolism in AF with creatinine clearance (CrCl) > 95 mL/min has been reported. The purpose of this meta-analysis is to further clarify the safety (major bleeding) and efficacy (stroke or systemic embolism) of edoxaban for AF patients with various CrCl. **Methods**: A systematic search of studies on edoxaban and warfarin in AF patients related to renal function was conducted in PubMed, Medline, Web of Science databases, EBSCO, Embase, and the Cochrane Central Register of Controlled Trials. In this meta-analysis (protocol number: PROSPERO CRD 42021245512), we included studies that provide specific data on three outcomes: ischemic stroke or systemic embolism (S/SE), bleeding, and all-cause mortality. **Results**: This meta-analysis enrolled two randomized controlled trials (RCTs) studies and two retrospective studies that enrolled 28,065 patients. According to CrCl, subjects are divided into three groups (CrCl 30–50 mL/min, CrCl 50–95 mL/min, CrCl > 95 mL/min). In AF patients with CrCl 30–50 mL/min, edoxaban 30 mg daily is similar to warfarin in the prevention of ischemic S/SE and all-cause mortality, resulting in lower bleeding rate and better net clinical outcome (ischemic S/SE: hazard ratio (HR), 0.85, 95% confidence interval (CI), 0.19–1.87; all-cause mortality: HR, 0.65, 95% CI, 0.35–1.19; bleeding: HR, 0.75, 95% CI, 0.60–0.93; net clinical outcome: HR, 0.75, 95% CI, 0.63–0.90). In the group of CrCl 50–95 mL/min, the net clinical outcome was more favorable with edoxaban 60 mg daily than warfarin (HR, 0.81, 95% CI: 0.68–0.96), and there was no significant difference between edoxaban 60 mg daily and warfarin in terms of prevention of bleeding, ischemic S/SE, and all-cause mortality. For AF patients with CrCl > 95 mL/min, there was a statistically significant difference in lower bleeding rate between edoxaban 60 mg daily and warfarin (bleeding: HR: 0.70, 95% CI: 0.58–0.84). There was no differential safety in ischemic S/SE, all-cause mortality, and net clinical outcome. **Conclusion**: Overall, edoxaban was superior to warfarin in terms of net clinical outcome in various groups of CrCl with AF patients. Although there was no significant difference in net clinical outcome between edoxaban and warfarin for AF patients with CrCl > 95 mL/min, edoxaban is not inferior to warfarin in safety and effectiveness in the various levels of CrCl. Edoxaban may be a more effective and safe treatment than warfarin for patients with chronic kidney disease (CKD) who require anticoagulation. More high-quality and long-term clinical research are needed to further estimate the effects of edoxaban.

## 1. Introduction

Atrial fibrillation (AF) is a type of supraventricular tachyarrhythmia accompanied by uncoordinated atrial electrical activity. AF leads to unsynchronized atrial contraction and irregular ventricular excitation [1] and may eventually cause death, stroke, heart failure, cognitive decline, depression, poor quality of life, and adverse prognosis, which places a tremendous burden on patients and the health care system worldwide. It is the most common persistent arrhythmia in adults, affecting over 33 million patients worldwide [2]; recent estimates report an AF prevalence of 2.7–7.4% among adults [3]. With aging populations and the strengthening of AF diagnosis, the prevalence rate gradually increases [4].

Recently, according to published studies, renal insufficiency has been an independent risk of stroke in patients with AF. Although the availability of novel oral anticoagulants (NOACs) has changed the landscape for the avoidance of AF-related systemic embolism and bleeding [5,6], few articles emphasize the role of edoxaban in the clinical benefit of patients with different degrees of renal function, especially in head-to-head comparison. This analysis is particularly meaningful in view of the US Food and Drug Administration (FDA) label that restricts use in patients with a creatinine clearance (CrCl) > 95 mL/min because of concerns of reduced relative efficacy in the prevention of stroke compared to warfarin [7].

Edoxaban is an oral anticoagulant that is a direct factor Xa inhibitor, which is removed by the kidney by about 50% [8]. In this meta-analysis, we included studies that could offer specific data on three outcomes: ischemic stroke or systemic embolism (S/SE), bleeding, and all-cause mortality. Our primary aim was to perform a systematic review of clinical research outcomes to explain the safety (bleeding) and efficacy (ischemic S/SE) of edoxaban in patients with AF at different levels of CrCl.

## 2. Materials and Methods

### 2.1. Data Sources and Searches

This systematic review and meta-analysis followed the preferred reporting items for systematic reviews and meta-analyses guidelines [9] (protocol number: PROSPERO CRD 42021245512). The aim of our study was to explore the effect of renal function in adults with edoxaban, compared with warfarin, on outcomes including ischemic stroke, bleeding, and all-cause mortality. PubMed, Medline, Web of Science databases, Cochrane Central Register of Controlled Trials, Embase, and EBSCO were searched from inception to April 2021 using the following keywords: “atrial fibrillation”, “edoxaban”, “warfarin”, “oral factor Xa inhibitor”, and “creatinine clearance”. No language limits were included. 

### 2.2. Selection Criteria

Two of the authors independently performed the literature search and extracted data from each qualified study. All studies fulfilled the following criteria: (1) randomized control trials (RCTs) or observational studies of clinically stable patients; (2) edoxaban compared with warfarin as anticoagulant treatment for patients with AF; (3) evaluation of renal function using CrCl at baseline, with a description of the number of patients; and (4) results reported according to CrCl. For clinical studies issued in more than one publication, the data were obtained from the most complete publication. The differences were determined by a third researcher.

### 2.3. Data Extraction and Quality Assessment

We performed the risk of bias tool for the RCTs and non-randomized studies of interventions (ROBINS-I) for retrospective studies using RevMan software (version 5.3; Cochrane Collaboration, Oxford, UK) [10]. The ROBINS-I has three subsets: pre-intervention, at-intervention, and post-intervention. Pre-intervention emphasizes bias as a result of confounding and bias in the selection of study participants. At-intervention emphasizes bias due to the classification of interventions. Post-intervention highlights bias in the deviations from intended interventions, in virtue of missing data, in measurement of outcomes, and in the selection of the reported result. Differences between the reviewers were discussed under the supervision of other authors.

### 2.4. Data Synthesis and Analysis

We assigned the results of every study as dichotomous frequency data. Statistical significance was set at *p* < 0.05. Hazard ratios (HRs) and 95% confidence intervals (CIs) were calculated. Data were collected and compared using a random effects model. Meta-regression was performed using weighted regression after log transformation of each HR value. Egger’s test was used to assess the potential publication bias [11]. Heterogeneity was estimated using the I^2^ statistic [12]. For the I^2^ statistic, a value of >50% was regarded statistically significant for heterogeneity. The software StataSE V.12.0 supported this analysis. Lastly, sensitivity analyses were performed using the leave-one-out approach.

## 3. Results

### 3.1. Study Search and Research Evaluation

We searched 4122 records identified through the databases without additional articles from other sources. A total of 1421 duplicate articles and 2697 incompatible titles, case reports, or abstracts were excluded. Finally, four unique records, which enrolled 28,065 patients, were included for full evaluation (Figure 1). The study characteristics are presented in Table 1. This meta-analysis involved two RCTs [8,13], and two retrospective studies, [14,15] which were evaluated using RevMan 5.3 and ROBINS-I [10]. respectively (Figure 2, Figure 3 and Figure 4).

### 3.2. Risk of Bias in Included Articles

The risk of bias outcome was determined following the Cochrane guidelines [10,16] and is summarized in Figure 2 and Figure 4, revealing a low risk for performance bias. The two retrospective studies [14,15] were included in the non-randomized group, which exhibited a low risk of bias in the classification of interventions and bias in the outcome data. They were assessed as a low risk of bias in the measurement of outcomes and were appraised as having low risk in the selection of the reported results. On the other hand, the two RCTs [8,13] reported a low risk of bias due to randomization procedures, differences from intended interventions, absent data, outcome measurement, and selection of the conveyed result.

### 3.3. Pooled Effect Estimates

#### 3.3.1. Safety Outcomes According to CrCl

According to CrCl, subjects are divided into three groups (CrCl 30–50 mL/min, CrCl 50–95 mL/min, CrCl > 95 mL/min). Overall, treatment with edoxaban was related to a significant decrease in the risk of bleeding compared with warfarin (HR, 0.76, 95% CI, 0.66–0.88). There was a significant difference in the risk of bleeding between the edoxaban 30mg daily and warfarin groups among patients with CrCl 30–50 mL/min (HR, 0.75, 95% CI, 0.60–0.93). In the group of CrCl > 95 mL/min, the use of edoxaban 60 mg daily was associated with a significant decrease in the risk of bleeding (HR, 0.70, 95% CI, 0.58–0.84). However, there was no significant difference in patients with 60 mg daily in the group of CrCl 50–95 mL/min (HR, 0.81, 95% CI, 0.61–1.06) (Figure 5).

#### 3.3.2. Efficacy Outcomes According to CrCl

Overall, there was no difference in the risk of ischemic stroke (HR, 0.84, 95% CI, 0.65–1.09). (Figure 6). The efficacy of edoxaban was similar to warfarin for the prevention of ischemic S/SE in the various levels of CrCl (CrCl 30–50 mL/min 30 mg daily: HR, 0.85, 95% CI, 0.19–1.87; CrCl 50–95 mL/min 60 mg daily: HR, 0.84, 95% CI: 0.63–1.11; CrCl > 95 mL/min 60 mg daily: HR, 0.86, 95% CI, 0.41–1.81).

Generally, no difference was observed in mortality reduction between edoxaban and warfarin in the various CrCl groups (CrCl 30–50 mL/min 30 mg daily: HR, 0.65, 95% CI, 0.35–1.19; CrCl > 50–95 mL/min 60 mg daily: HR, 0.46, 95% CI, 0.16–1.33; CrCl > 95 mL/min 60 mg daily: HR, 1.12; 95% CI, 0.87–1.44) (Figure 7).

On the whole, there was significance in net clinical outcome between edoxaban and warfarin (HR, 0.80, 95% CI, 0.72–0.89). The primary clinical outcomes of S/SE, major bleeding, and all-cause death was more favorable for edoxaban compared with warfarin across the range of renal function subgroups (CrCl 30–50 mL/min 30 mg daily: HR, 0.75, 95% CI, 0.63–0.90; CrCl > 50–95 mL/min 60 mg daily: HR, 0.81, 95% CI, 0.68–0.96). In the group of CrCl > 95 mL/min 60 mg daily, there was no difference in the net clinical outcome (HR, 0.83, 95% CI, 0.68–1.02) (Figure 8).

In this meta-analysis on the described efficacy and safety outcomes of edoxaban, there was seemingly no potential bias upon the sensitivity analysis (Appendix A). These figures are shown in Appendix A.

## 4. Discussion

In view of numerous studies evaluating the effect of NOACs for preventing stroke or systemic embolism, few articles placed emphasis on the effects of NOACs for preventing stroke or systemic embolism and major bleeding in patients with different degrees of renal function, especially for head-to-head comparisons. To the best of our knowledge, this is the first meta-analysis to investigate the effectiveness and safety of edoxaban and warfarin in patients with AF. We enrolled four studies based on the selection criteria, prevailing biases, quality of data, and defined outcomes (Table 1) [8,13,14,15]. The main findings of our study are as follows: (1) treatment with edoxaban in relation to renal function was correlated with a significant reduction in the risk of bleeding, ischemic stroke, and mortality compared to warfarin; (2) edoxaban is not inferior to warfarin in terms of the risk of bleeding, ischemic stroke, and mortality in AF patients with supernormal CrCl (>95 mL/min); (3) for subjects with CKD who need anticoagulation, edoxaban may be a more effective and safe treatment than warfarin.

In patents with CrCl 30–50 mL/min, edoxaban 30 mg daily was considered as the higher-dose edoxaban regimen [8]. We found that edoxaban 30 mg daily is similar to warfarin in the prevention of ischemic S/SE and all-cause mortality, resulting in lower bleeding rates and a better net clinical outcome for AF patients with CrCl 30–50 mL/min (ischemic S/SE: HR, 0.85, 95% CI, 0.19–1.87; bleeding: HR, 0.75, 95% CI, 0.60–0.93; all-cause mortality: HR, 0.65, 95% CI, 0.35–1.19; net clinical outcome: HR, 0.75, 95% CI, 0.63–0.90). In a meta-analysis, for AF patients CrCl 30–50 mL/min, edoxaban 30 mg daily was beneficial in all non-vitamin K oral anticoagulants comparisons for safety. For the efficacy, edoxaban 30 mg daily is also superior to warfarin and rivaroxaban [17]. This is consistent with our results.

For AF patients with CrCl 50–95 mL/min, edoxaban 60 mg daily is better than warfarin in net clinical outcome in the AF patients with CrCl 50–95 mL/min (HR, 0.81, 95% CI, 0.68–0.96). In terms of prevention of bleeding, ischemic S/SE, and mortality, there was no significant difference between edoxaban 60 mg daily and warfarin (bleeding: HR: 0.81, 95% CI, 0.61–1.06; ischemic S/SE: HR, 0.84, 95% CI, 0.63–1.11; all-cause mortality: HR, 0.46, 95% CI, 0.16–1.33). These conclusions are consistent between edoxaban 60 mg daily in AF Patients with CrCl 50–95 mL/min and edoxaban 30 mg daily in subgroup of CrCl 30–50 mL/min. Decreased renal clearance result in higher drug levels, because edoxaban is renally cleared at 50% [18].

For patients with CrCl > 95 mL/min, there was a statistically significant difference in a lower bleeding rate between edoxaban 60 mg daily and warfarin (bleeding: HR: 0.70, 95% CI, 0.58–0.84). There was no differential safety in ischemic S/SE, all-cause mortality and net clinical outcome (ischemic S/SE: HR, 0.86, 95% CI, 0.41–1.81; all-cause mortality: HR, 1.12, 95% CI, 0.87–1.44; net clinical outcome: HR, 0.83, 95% CI, 0.68–1.02). The FDA released a warning concerning treatment with edoxaban in patients with creatinine clearance (CrCl) > 95 mL/min [7]. However, we found a nonsignificant trend towards a reduced risk of all-cause mortality, ischemic S/SE and net clinical outcome between edoxaban 60 mg daily and warfarin. In AF patients with CrCl > 95 mL/min, edoxaban 60 mg daily is not inferior to warfarin in safety and effectiveness. This is consistent with So Young Lee’s research [14].

For over half a century, warfarin has been regarded as the standard anticoagulant treatment to reduce the risk of ischemic stroke in patients with valvular and non-valvular AF. However, the toxic dosage of warfarin is proximal to the required dose in order to achieve a pharmaceutical effect. The warfarin dosage response is associated with environmental, demographic, clinical, and genetic factors [19]. Edoxaban effectively inhibits the conversion of prothrombin to thrombin, thus reducing thrombus formation and promoting other advantages. Currently, it has been recommended for the treatment of venous thromboembolism and AF [20]. Edoxaban is cleared through the kidney regardless of renal function. The FDA issued a warning on taking edoxaban for patients with CrCl > 95 mL/min because of a reported decrease in efficacy for the prevention of systemic embolism compared with warfarin [7].

Renal impairment often increases the risk of bleeding and thromboembolism among patients with chronic kidney disease (CKD) [21]. The mechanism is probably related to damaged platelet adhesion and aggregation, altered endothelium, activated coagulation pathway, and reduced blood coagulation inhibitors [22]. Furthermore, worsening renal clearance consequently increases the risk of bleeding and decreases the clearance of anticoagulants. Factor Xa inhibitors have been included in the routine treatment for patients with mild-to-moderate CKD [23]. In a case of a 75-year-old woman with mechanical mitral valve replacement who remained stable without taking anticoagulant medication since the operation, we found that the patient suffered from factor X deficiency and published the European Heart Journal Case Reports [24]. We found that the primary clinical outcomes of S/SE, major bleeding, and all-cause death was more favorable for edoxaban compared with warfarin for AF Patients with CrCl 50–95 mL/min. Therefore, edoxaban may be a more effective and safe treatment than warfarin for patients with CKD who require anticoagulation.

The efficacy and safety outcomes of edoxaban in comparison with warfarin for stroke prevention in patients with AF have been consistently described in NOAC treatment for non-valvular AF and venous thromboembolism in RCTs [25,26,27,28,29]. Additionally, Zou R et al. [30] published a meta-analysis of five RCTs, including 72,608 patients, and compared the clinical efficacy between the use of NOACs and warfarin for non-valvular AF in relation to the different levels of renal function. This review stated that NOACs had a better clinical benefit than warfarin in varying degrees of renal function. Our study reported that edoxaban was correlated with a significantly lower risk of bleeding, ischemic stroke, and mortality in AF patients with respect to CrCl.

### Study Limitations

Our study has some limitations that merit further consideration. First, some studies in relation to kidney function were not presented, and no hypothesis was stated in the entire study population. Second, warfarin and edoxaban head-to-head trials on renal function are still lacking. Third, we performed the Cockcroft–Gault equation, which could have introduced selection bias, as it is partial in body weight. Consequently, we divided the patients based on the nearest CrCl cut-off point, which could have introduced sampling bias. Fourth, the included studies were post-hoc analyses of RCTs and retrospective studies that may have introduced other biases. In particular, several studies did not clearly define their clinical endpoints, especially bleeding subtype (major or minor), stroke subtype, and stroke or systemic embolism.

## 5. Conclusions

Our systematic review and meta-analysis showed that edoxaban use, compared to warfarin, is related to a decreased risk of ischemic stroke and mortality, regardless of the patient’s renal function. Therefore, edoxaban may be a more effective and safer treatment for patients with CKD who require anticoagulation than warfarin. In view of the two RCT studies included this study, more RCT studies need to be carried out to prove the efficacy and safety of edoxaban in patients with renal insufficiency in the future.

## Figures and Tables

**Figure 1 medicines-10-00013-f001:**
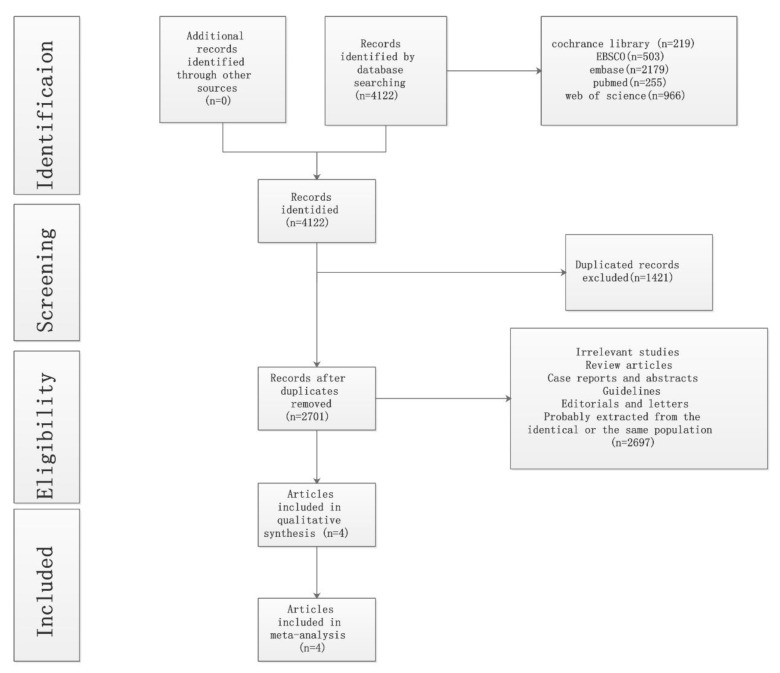
Preferred reporting items for systematic reviews and meta-analyses (PRISMA) flow chart.

**Figure 2 medicines-10-00013-f002:**
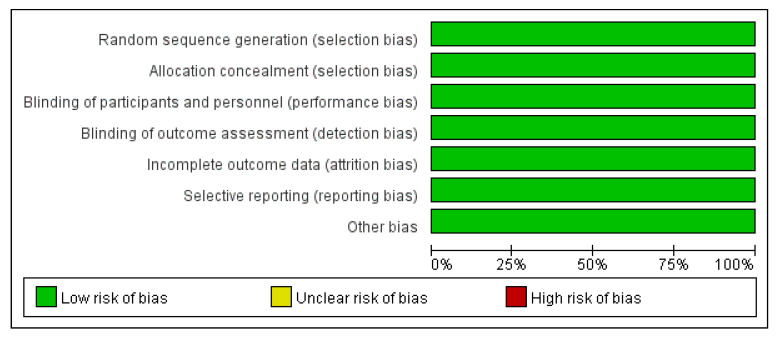
Summary of each RCT appraised by Revman 5.3.

**Figure 3 medicines-10-00013-f003:**
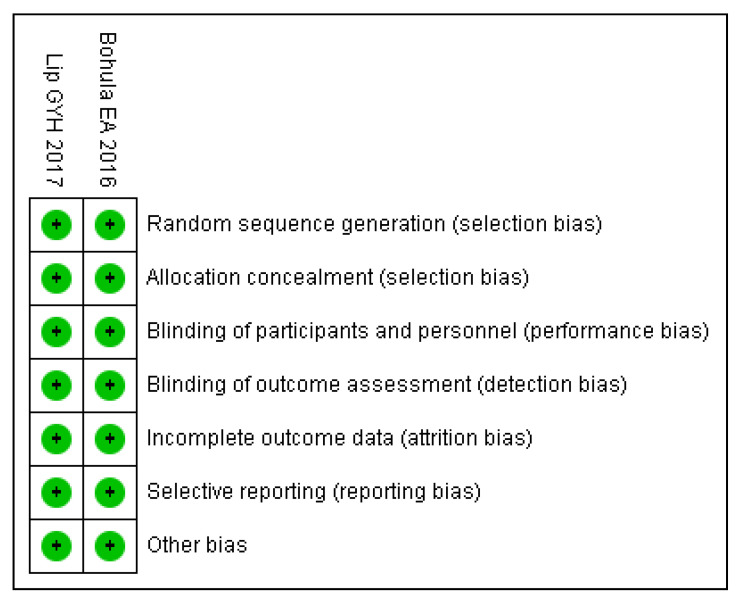
Risk of bias graph of each RCT appraised by Revman 5.3.

**Figure 4 medicines-10-00013-f004:**
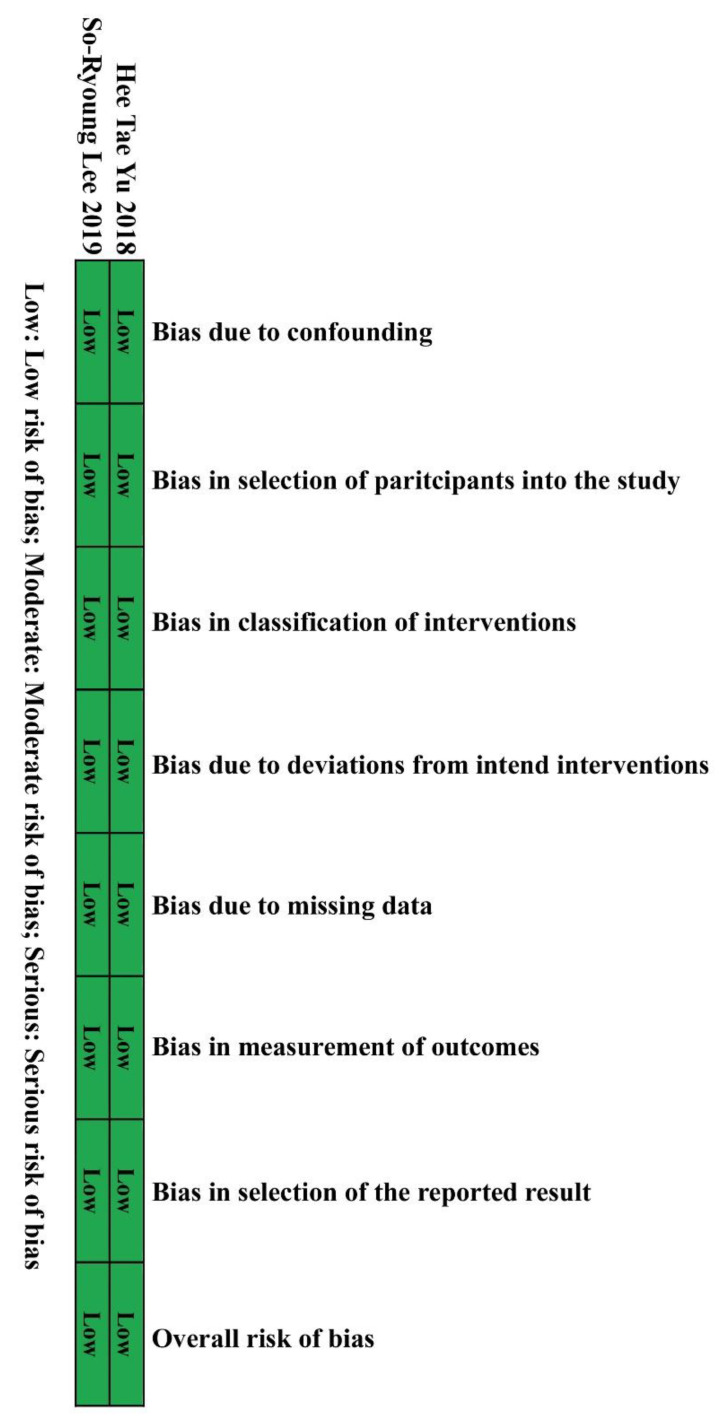
Summary of each retrospective study evaluated using ROBINS-I.

**Figure 5 medicines-10-00013-f005:**
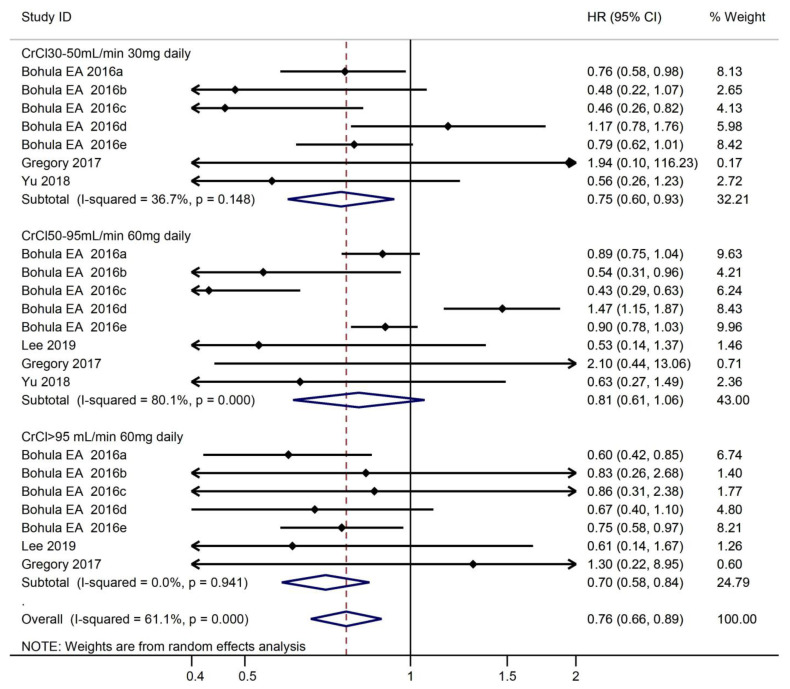
Risk of bleeding and use of edoxaban versus warfarin in relation to CrCl.

**Figure 6 medicines-10-00013-f006:**
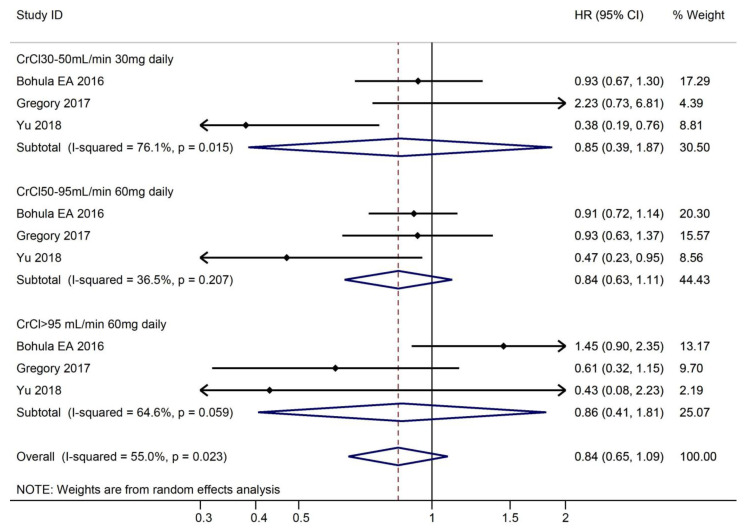
Risk of ischemic stroke and use of edoxaban versus warfarin in relation to CrCl.

**Figure 7 medicines-10-00013-f007:**
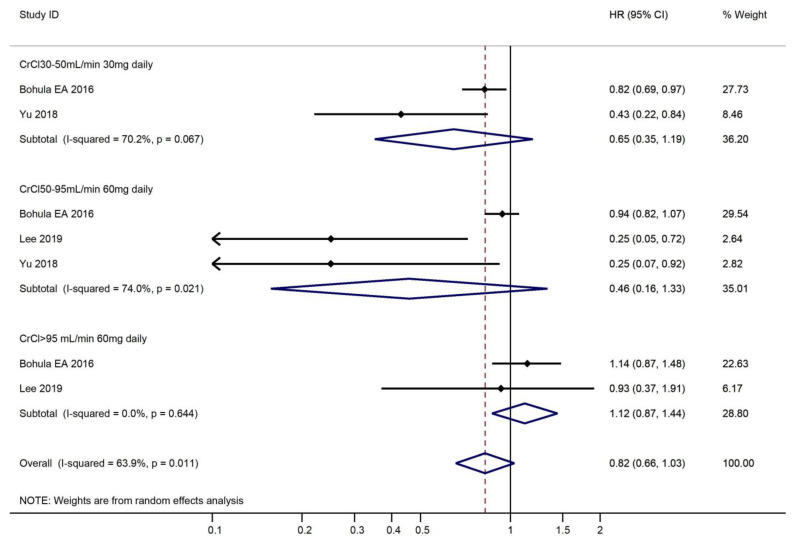
Risk of mortality and use of edoxaban versus warfarin in relation to CrCl.

**Figure 8 medicines-10-00013-f008:**
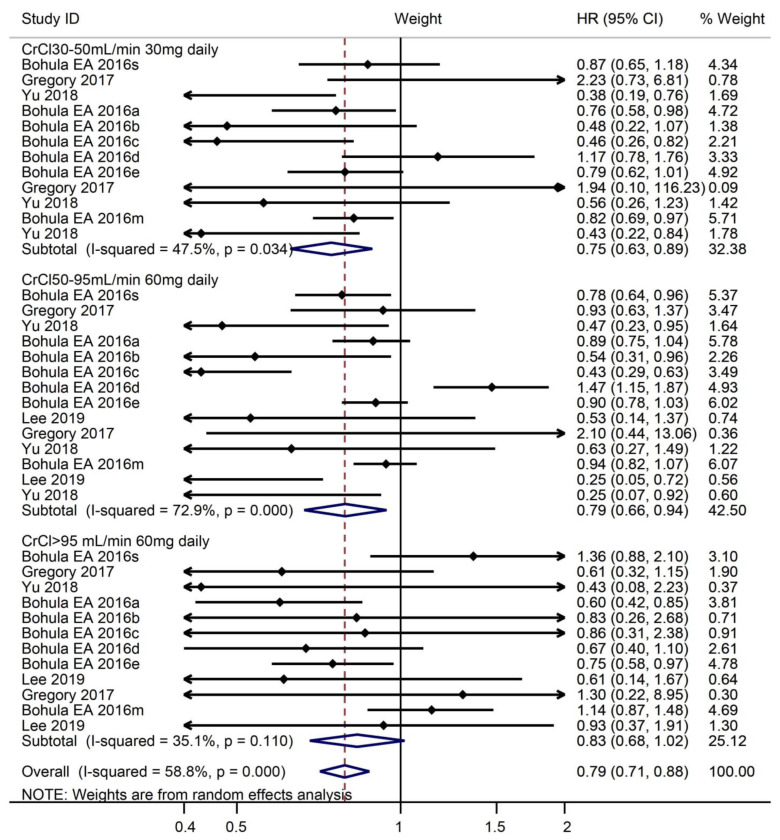
Primary efficacy, safety, and net clinical outcome end points by prespecified CrCl subgroups.

**Table 1 medicines-10-00013-t001:** Summary of the retrieved articles on edoxaban versus warfarin among AF patients in relation to CrCl.

Characteristics	Trial			
	2016 Bohula, E.A.	2017 Lip, Gregory	2018 Hee Tae	2019 So-Ryoung Lee
Country	International	International	Korea	Korea
Design	Multinational, randomized, double-blind	RCT	Retrospectively	Retrospective nationwide cohort study
Registry	NCT00781391	NCT02072434	NA	NCT02786095
Number of Patients	14,071	1095	11,712	11,071
Endpoints/second outcome	Endpoints: stroke or systemic embolism major bleeding all-cause death; Additional safety end points: intracranial hemorrhage, gastrointestinal bleeding, minor bleeding.	Endpoints: stroke, systemic embolic event, myocardial infarction, any bleeding, cardiovascular death.	Endpoints: stroke or systemic embolism, major bleeding, and death from any cause; Secondary outcomes: intracranial bleeding, gastrointestinal bleeding, myocardial infarction, or admission for heart failure.	Endpoints: ischemic stroke, major bleeding, all-cause death.
CrCl, mL/min	30–50, 50–95, >95	15–30, 30–50, 50–80, 80–95, ≥95	30–50, 50–70, 70–95, >95,	80–95, >95
Follow Up	2.8 (interquartile range, 2.4–3.2 years)	28 days	5.0 months (interquartile range, 2–7 months)	1.2 years (interquartile range, 0.6–1.9 years)
CHADS2 risk score	2.8	2.6	4.2 ± 1.7	3.0 ± 1.6
Medical history	Edoxaban 30 mg vs. Edoxaban 60 mg vs. Warfarin Diabetes: 1:1:1 Hypertension: 1:1:1 Heart failure: 1:1:1 Ischaemic stroke or transient ischaemic attack: 1:1:1 Previous VKA used: 1:1:1	Edoxaban vs. Warfarin Congestive heart failure:1:1 Coronary artery disease: 1:1 Hypertension: 1:1 Diabetes: 1:1 Ischaemic heart disease: 1:1 Ischaemic stroke or transient ischaemic attack: 1:1 Life-threatening bleed: 1:1	Edoxaban 30 mg vs. edoxaban 60 mg vs. Warfarin Diabetes: 1:1:1 Hypertension: 1:1:1 Heart failure: 1:1:1 Ischaemic stroke or transient ischaemic attack: 1:1:1 Vascular disease: 1:1:1 Dyslipidemia: 1:1:1	Edoxaban vs. Warfarin Heart failure: 1:1 Dyslipidemia: 1:1 Hypertension: 1:1 Diabetes: 1:1 Myocardial infarction: 1:1 Peripheral artery disease: 1:1 Chronic obstructive pulmonary disease: 1:1

## Data Availability

Data available on request from the authors.

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
