# Peer review of "Efficacy and Safety of Renal Function on Edoxaban Versus Warfarin for Atrial Fibrillation: A Systematic Review and Meta-Analysis"

_medicines, 2023, doi:10.3390/medicines10010013_

Round 1

Reviewer 1 Report

The authors have investigated and important cardiovascular theme regarding NOAC. Edoxaban thanks to its easy to use therapeutic scheme is one of the most used NOAC world wide. Its limitation in data sheet is patients with CrCl > 95 ml/min. This review and meta-analysis highlights how Edoxaban is safer than Warfarin regardless of CrCl even in supernormal filtration patients (CrCl >95 mL/min). 

The study is interesting, well thought of and overall well written. Introduction is sufficient and on point. Materials and Methods are described in an according manner. Results are of universal interpretation. Discussion in on target. 

My minor revision consists in the need in the conclusion for future prospective. This review should set for a change in drug leaflet and guidelines. 

Author Response

We would like to thank you for your careful reading, helpful comments, and constructive suggestions, which has significantly improved the presentation of our manuscript.

We have carefully considered all comments from the reviewers and revised our manuscript accordingly. In the following section, we summarize our responses to each comment from the reviewers. We believe that our responses have well addressed all concerns from the reviewers. We hope our revised manuscript can be accepted for publication.

Response to Reviewer 1 Comments

Point 1: My minor revision consists in the need in the conclusion for future prospective.

Response 1: In view of the two RCT studies included this study, more RCT studies need to be carried out to prove the efficacy and safety of edoxaban in patients with renal insufficiency in the future.

We deeply appreciate your consideration of our manuscript, and we look forward to receiving comments from you. If you have any queries, please don’t hesitate to contact me at the address below.

Thank you and best regards!

Yours sincerely, Dong-jin Wang

Corresponding Prof. Dong-jin Wang, E-mail: [email protected].

Reviewer 2 Report

I read with great interest the manuscript titled “Efficacy and Safety of Renal Function on Edoxaban versus Warfarin for Atrial Fibrillation: A Systematic Review and Meta-analysis” by Wang et al. The authors conclude that Edoxaban is associated with decreased risk of bleeding, stroke/systemic embolism, and all-cause mortality regardless of creatinine clearance. They also conclude that Edoxaban is not inferior to warfarin in terms of the risk of bleeding, ischemic stroke, and all-cause mortality in AF patients with CrCl>95 ml/min.

They have included 2 RCTs including the ENGAGE AF-TIMI 48 trial which is the basis of the FDA black box warning to not use Edoxaban in patients with CrCl>95 ml/min. In addition, 2 observational studies from Korea were also included and notably, all NOACs (including Edoxaban) have shown to be more effective in the Asian population at preventing ischemic strokes and decreasing the risk of intracranial hemorrhage in comparison to warfarin. Observational studies are not as robust as RCT due to inherent design limitations, but both observational studies included have performed propensity matching that limits the amount of variation in the study population though residual confounding cannot be completely ruled out. It is important that authors elaborate on the effect of NOACs, particularly edoxaban, in comparison to warfarin in the Asian population. They should also mention the limitations of including observational studies in the meta-analysis.

In the ENGAGE AF-TIMI 48 study, the patients in the warfarin group had a much lower rate of ischemic stroke in comparison to other NOAC trials and the patients in the warfarin group had greater than average time in therapeutic range, which is much higher than seen in real life, that could have contributed to their results. In European Union, Edoxaban can be used in patients with CrCl >95 ml/min and there is no black box warning against their use in this population.

It is also important to elaborate on the duration of follow-up in various studies included in the limitation section. The ENGAGE-AF TIMI 48 trial followed the patients for a median of close to 3 years which make their results more robust.

In spite of the limitations listed above, this study adds to the available evidence that Edoxaban might be safe to use in patients with CrCl >95 ml/min, especially in carefully selected patients of Asian race.

Minor comments:

1.      In line 45, change “not ” to “no significant difference”.

2.      In line 181, please add Edoxaban to the sentence “there was significance in net clinical outcome between and warfarin”.

3.      Please correct “valvar” to “valvular”.

Author Response

We would like to thank you for your careful reading, helpful comments, and constructive suggestions, which has significantly improved the presentation of our manuscript.

We have carefully considered all comments from the reviewers and revised our manuscript accordingly. In the following section, we summarize our responses to each comment from the reviewers. We believe that our responses have well addressed all concerns from the reviewers. We hope our revised manuscript can be accepted for publication.

Response to Reviewer 2 Comments

Point 1: In line 45, change “not ” to “no significant difference”.

Response 1: Although there was no significant difference in net clinical outcome between edoxaban and warfarin for AF patients with CrCl >95 ml/min, edoxaban is not inferior to warfarin in safety and effectiveness in the various levels of CrCl.

Point 2: In line 181, please add Edoxaban to the sentence “there was significance in net clinical outcome between and warfarin”.

Response 2: On the whole, there was significance in net clinical outcome between edoxaban and warfarin (HR, 0.80, 95%CI, 0.72-0.89).

Point 3:  Please correct “valvar” to “valvular”.

Response 3: I have done it.

We deeply appreciate your consideration of our manuscript, and we look forward to receiving comments from you. If you have any queries, please don’t hesitate to contact me at the address below.
